# Rapid Microfluidic Mixer Based on Ferrofluid and Integrated Microscale NdFeB-PDMS Magnet

**DOI:** 10.3390/mi11010029

**Published:** 2019-12-25

**Authors:** Ran Zhou, Athira N. Surendran, Marcel Mejulu, Yang Lin

**Affiliations:** 1Department of Mechanical and Civil Engineering, Purdue University Northwest, Hammond, IN 46323, USA; nair23@pnw.edu (A.N.S.); mmejulu@pnw.edu (M.M.); 2Department of Mechanical and Industrial Engineering, University of Illinois at Chicago, Chicago, IL 60607, USA; ylin212@uic.edu

**Keywords:** micromixer, microfluidics, ferrofluid, magnetic field, neodymium (NdFeB) powders

## Abstract

Ferrofluid-based micromixers have been widely used for a myriad of microfluidic industrial applications in biochemical engineering, food processing, and detection/analytical processes. However, complete mixing in micromixers is extremely time-consuming and requires very long microchannels due to laminar flow. In this paper, we developed an effective and low-cost microfluidic device integrated with microscale magnets manufactured with neodymium (NdFeB) powders and polydimethylsiloxane (PDMS) to achieve rapid micromixing between ferrofluid and buffer flow. Experiments were conducted systematically to investigate the effect of flow rate, concentration of the ferrofluid, and micromagnet NdFeB:PDMS mass ratio on the mixing performance. It was found that mixing is more efficient with lower total flow rates and higher ferrofluid concentration, which generate greater magnetic forces acting on both streamwise and lateral directions to increase the intermixing of the fluids within a longer residence time. Numerical models were also developed to simulate the mixing process in the microchannel under the same conditions and the simulation results indicated excellent agreements with the experimental data on mixing performance. Combining experimental measurements and numerical simulations, this study demonstrates a simple yet effective method to realize rapid mixing for lab-on-chip systems.

## 1. Introduction

Microfluidics is the study pertaining the design and manufacturing of small devices that embody tiny channels to enable precise control and manipulation of fluids at the micro scale, typically ten to hundreds of microns [1,2]. In recent years, microfluidic devices have become increasingly more popular as evidenced by numerous applications in the fields of biology [3,4], drug delivery and screening [5,6], clinical diagnostics [7,8], and other disciplines. Mixing of reagents and analytes is one of the most important applications and an essential step in microfluidic devices for pre-processing, dilution, or inducing reactions between samples and reagent [9]. Almost every chemical assay requires mixing of two or more fluids or reagents with a sample and the mixing on a microscale is defined as micromixing. In addition, rapid mixing is often required to avoid flocculation and sedimentation. Therefore, the ability to rapidly mix liquids in microscale greatly improves the performance of microfluidic systems. However, due to the small channel size in microfluidic devices, fluid flow in a microchannel stays laminar, with a very low characteristic Reynolds number (Re<1). The complete and homogenous mixing of fluids is mainly dominated by the slow process of molecular diffusion. Consequently, fluids containing molecules such as proteins and DNAs with small diffusion coefficients require longer channels which in turn takes a longer time for complete mixing [10,11]. This gives rise to the need for rapid mixing in microfluidic devices, especially those involving fluids of large molecules, to save mixing time, shorten the length of channels, and ensure complete mixing [12,13]. 

Based on the studies and applications done in previous years, mixing in microfluidics can be classified into passive and active mixing [14]. In the passive method, mixing is induced by driving fluids through channels and the hydrodynamic effects with appropriate geometric designs. Although this type of method does not require additional energy except that for driving fluids, it usually requires a longer channel and therefore takes a longer time to achieve homogenous mixing [15]. In contrast to passive mixing, active mixing uses external forces to accelerate mixing. With a high mixing efficiency, active mixing does not require long channels or increased time of contact to achieve mixing. Active micromixers rely on external perturbations caused by forces such as acoustic, electrical, magnetic, and thermal forces to induce mixing [14,15,16]. These methods have been used extensively in different applications by numerous researchers of microfluidics. Deshmukh et al. [17] reported a fast mixing method driven by pulsatile flow micropumps. The micromixer is made of Silicon on Insulator (SOI) and bonded quartz wafers. The theory of mixing process is based on using a heater made of polysilicon resistors on quartz to generate bubbles that drive the fluids into the channel and push the fluid. The velocity gradient observed in the downstream channel caused a distortion of the bulge and it was observed that the method increased the mixing interface and mixing efficiency. This work also modeled the device numerically using computational fluid dynamics (CFD) with water and with glucose to validate the mixing performance. Fuji et al. [18] proposed a plug and play lab on a chip system that consists of a part that drives the fluids and polydimethylsiloxane (PDMS) fluidic channel. Their work offers a means to increase mixing efficiency using generated pressure disturbance which is provided by the alternate pulsed flow operation. El Moctar et al. [19] developed a microfluidic platform using electrohydrodynamic force to mix two fluids with different electrical properties. The mixing was generated by an electrical field that is perpendicular to the flow interface because of the resulting electrohydrodynamic force. The effect of frequency and the type of current on the mixing efficiency was discussed and the force intensity was found to decrease if the frequency was higher than a certain range. Samei et al. [20] reported a rapid mixing method by using high frequency voltages to manipulate the droplet. In their study, it was found that the mixing efficiency increased when the frequency and voltages increased, but the frequency was required to be set at an effective range to enable high-performance mixing. However, the heat generated by the high frequency actuation was detected which could be harmful for biological samples. Ober et al. [21] proposed a rotating impeller and its rapid motion enabled efficient mixing. This technique was applied to develop a nozzle printhead used for 3D printing techniques. However, these methods often lead to temperature rises in the system and can potentially cause damage to cells due to the resulting high energy. 

Another approach to improve mixing efficiency is to use external magnetic fields. Lu et al. [22] presented a magnetic bar that can be controlled by an external rotating magnetic field. The rapid stirring of the magnetic bar inside the fluid is able to generate bulk motion and thus mix the flows inside the channel. Micromachining and micromolding techniques were used for the fabrication of this device. This stirring method has been proved to reduce the mixing time and improve the complete mixing performance. Ferrofluids have received much attention recently to enhance the mixing efficiency between other sample fluids due to its characteristics of biological compatibility, thereby it can be applied widely in biological sensors to mix proteins, DNAs, and other blood cells [23,24,25,26]. The ferrofluid becomes strongly magnetized by an external magnetic field, and the rapid mixing between ferrofluid and other biological reagent solutions can be achieved in microchannels with reduced length and flow time. For real applications in the biological microfluidic field, use of ferrofluid for mixing enhancement usually serves as a pre-processed “tagging” step, followed by a second step for selective isolation of bio-entities [24,25,26]. For instance, in order to detect and isolate the pathogen from the original biofluid, ferrofluid is injected at one of the Y-shaped microchannel inlets to mix with the biofluid from the other inlet. With externally micro-magnetos, rapid mixing occurs, and magnetic nano-particles are distributed throughout the mixing fluids and preferentially bond with pathogens (due to its specific morphology). Once tagged with magnetic nano-particles, they become targeted pathogens that can be further detected and isolated in successive procedures [24]. Mao et al. [26] designed and developed a mixing prototype integrated with angled electrodes which can cause local vortices to generate mixing. Water-based ferrofluids can be controlled and manipulated by the alternating magnetic fields. The effect of frequency on the mixing performance was also discussed in their work. Nouri et al. [27] studied the mixing of water and ferrofluid based on the magnetic field generated by a permanent magnet. The effects of flow rate and concentration on the mixing performance were discussed from both experimental and simulation sides. However, these traditional magnetic methods rely on bulky permanent magnets or electromagnets, so it is difficult to control the mixing performance very precisely and consistently or the generated heat by electrodes is very harmful for biological samples.

In this study, we proposed a low-cost yet efficient microfluidic device containing a microfluidic channel and microscale magnets for the rapid mixing between ferrofluid and buffer flow. The microfluidic device is fabricated using the soft lithography method. The microscale magnet is made of a mixture of neodymium (NdFeB) powders and polydimethylsiloxane (PDMS) and is located only 150 µm away from the microfluidic channel as in Figure 1a. The microscale magnets generate relatively strong magnetic forces acting on the magnetic nanoparticles in the ferrofluid to enable the rapid mixing of ferrofluid and buffer flow more effectively. Ferrofluid and distilled water (buffer) were injected into the fluidic channel and the interaction of the fluids was observed at different positions along the microfluidic channel. It was found that the mixing performance is more efficient when the total flow rate is lower, because the smaller total flow rate provides more residence time for magnetic force to act on the ferrofluid to be mixed with water thoroughly. It was also observed that the mixing efficiency improves as the ferrofluid concentration increases, because increasing the concentration of ferrofluid results in greater magnetic force which in turn improves the mixing efficiency. Numerical simulations were performed to validate the mixing performance in the channels and a close match is found between the experimental measurements and simulation results. This study demonstrates a simple yet effective method for the rapid mixing processes between ferrofluid and buffer by the integrating the microscale permanent magnets into microfluidic devices.

## 2. Work Concept and Materials

### 2.1. Work Concept

In this work, we proposed a low-cost and simple method to realize the rapid mixing between ferrofluid and distilled water using embedded permanent microscale magnets. The microscale magnet is fabricated 150 µm away from the fluidic channel (Figure 1a) and thus able to generate a strong magnetic field and gradient to agitate the mixing of ferrofluid and water effectively. The width and height of the microscale magnet is w=500 μm and h=1000 μm, and the gap between each micromagnet bar is g=500 μm. $h^{\prime }=500\;\mu m$ is the width of connected microbar. This design has been proved to be able to generate optimal magnetic field and its gradients. The length L of both microfluidic channel and micromagnet is 20 mm. Figure 1b is the enlarged sketch of the microfluidic channel near the fluid entrances. The width of the microfluidic channel is $W_{c}\;=150\;\mu m$. The microfluidic channel has two fluidic inlets: the ferrofluid (ENG 408, Ferrotec, Santa Clara, CA, USA) is injected through Inlet 1 and the buffer flow with distilled water is injected at Inlet 2. From Figure 1c, the depth of the microfluidic channel and magnetic microstructures is 35 μm, which is the depth of dry photoresist film (MM540, 35 μm thick, DuPont, Wilmington, DE, USA). The microfluidic device in Figure 1d is fabricated by using the soft lithography method. The microscale magnet is made of a mixture of neodymium (NdFeB) powders and polydimethylsiloxane (PDMS) and located beside the microfluidic channel. The mixture was permanently magnetized by an impulse magnetizer. With the magnetic field and its gradient (Figure 1e) generated by microscale magnets, the mixing between ferrofluid and distilled water is achieved at the outlet of the fluidic channel because of the magnetization of the ferrofluid. The magnetic field is simulated by finite element method magnetics (FEMM) [28]. The magnetic coercivity of the microscale magnet was determined from experimental data, with $H_{c}$ being approximately 94,000 A/m, and the rectangular shape has been proven to be the optimal design to generate strong magnetic field and its gradients in previous works [29].

### 2.2. Microfluidic Device Fabrication

Figure 2 illustrates the fabrication steps of the microfluidic device with embedded magnets. A thin layer of dry photoresist film was laminated on a 2” by 3” copper plate at 100 °C using a thermal laminator. The laminated copper plate was then exposed to ultraviolet (UV) light in a dark room for 30 s through a transparent photo mask (10,000 dpi, CAD/Art Services, Bandon, OR, USA) with a custom design. After the exposure, a film was developed using a solution comprising of 1 L distilled water and 10 g Na_2_CO_3_ at a temperature of 35 °C. After the development, the film was rinsed in distilled water to stop further reactions, and dried to make the copper master mold. The PDMS base and initiator were thoroughly mixed with the mass ratio 10:1 respectively, degassed, and then poured onto the copper master mold. The cast was then cured in the oven at a temperature of 70 °C for 2 h, after which the PDMS was peeled off from the master mold. Excess material was cut off, and holes were punched in the PDMS to create channel inlets and outlets. The PDMS was cleaned using isopropanol (IPA), rinsed with distilled water, subjected to corona surface treatment (N001-020, UV Process Supply, Inc, Chicago, IL, USA), and then bonded with a glass microscope slide. Following, NdFeB powders (MQFP-B-20076, Molycorp Magnequench, Singapore) were thoroughly mixed with a premixed liquid PDMS at a weight to weight ratio of 2:1 respectively. The mixture was degassed then injected into the microstructure chamber of the microfluidic device to create the micromagnet. After injection, the microfluidic device was heated on a hotplate to cure the NdFeB-PDMS mixture. The fast curing process prevents the coagulation and sedimentation of NdFeB powders. The microfluidic device was heated up again in an oven overnight to ensure the complete curing of the mixture of NdFeB and PDMS. 

The bonding performance between the microfluidic channel and glass slide will also be consolidated during the reheating process. When the curing process was finished, the microfluidic device was placed in the magnetization chamber of an impulse magnetizer (IM 10, ASC Scientific, Narragansett, RI, USA), making the solid NdFeB-PDMS mixture permanently magnetized.

### 2.3. Experiment Setup and Materials

As can be seen from Figure 3, the flow rate at each inlet of the microfluidic channel was controlled by a syringe pump (Cole Parmer/KD Scientific 74900, Holliston, MA, USA) (Figure 3b). The microfluidic device was mounted on an inverted Amscope IN300TC-FL microscope stage. The observed flow phenomena were magnified by a digital microscope and recorded by a Photron AX100 high-speed camera (Photron, Tokyo, Japan). The working fluid used in this experiment is water-based ferrofluid EMG 408 whose magnetic nanoparticle concentration is 1.2% (v/v), dynamic viscosity is μ=2 mPa·s and the magnet susceptibility is$\;\chi_{f}=0.5$ according to the specs from Ferrotec (USA) Corporation. In our experiment, the initial concentration and properties was used as base experiment, and the original ferrofluid was diluted to 0.4% (v/v) with distilled water in the compared experiment to discuss the effect of ferrofluid concentration. The ferrofluid was injected into the upper inlet, and distilled water was injected into lower inlet as the buffer solution.

## 3. Numerical Simulation 

Computational fluid dynamics (CFD) simulations are performed to investigate microscale fluid flow and mixing in the channel used in this work with and without the effect of externally imposed magnets (Details of numerical modeling schemes are in the Appendix A). For the throughput range of 0.3–1.2 mL/h, the micro-channel fluid flow is well within the laminar regime (Re~O(0.1)), which reaches a steady-state condition within seconds during experiments. The governing equations for incompressible steady-state laminar flows therefore include the continuity and momentum [27]:(1)∇·u = 0
(2)$$\rho \left(u\cdot \nabla u\right)=-\nabla P+\eta \nabla^{2}u+F_{m}$$
where u is the fluid velocity (m/s), *P* is pressure (Pa), ρ is density ($\mathrm{kg}/m^{3}$), and η is the dynamic viscosity (Pa·s) of the fluid, and $F_{m}$ is the magnetic force acting on the ($N/m^{3}$). Buoyancy, gravitational forces and interaction forces between particles are neglected in the current simulations thanks to the diluted nanoparticle content in the ferrofluid.

Besides modeling fluid flow, the distribution of ferrofluid concentration in the microchannel is also obtained via the solution of a steady-state advection-diffusion equation, which also characterizes the mixing performance with the microchannel and magnets configurations: (3)$$\left(u\cdot \nabla \right)C=D\nabla^{2}C$$
where *C* represents the ferrofluid concentration (mole/$m^{3})$ and *D* is the mass diffusivity ($m^{2}/s)\;$between ferrofluid and water. Despite that the equation system (Equations (1)–(3)) appears deceptively straight forward that a simple one-way coupling would seemingly fit the solution with *C* transported as a passive scalar, the microchannel mixing process is truly a multi-physics phenomenon with the fields strongly coupled with each other. In cases without the external magnets, the density and dynamic viscosity of the mixture fluid are directly related to the local ferrofluid concentration through equations below [27,30]: (4)$$\rho_{\mathrm{mix}}=C\rho_{f}+\rho_{w}$$
(5)$$\eta_{\mathrm{mix}}=\eta_{f}e^{R\left(1-C\right)}$$
where *R = ln(*$\frac{\eta_{w}}{\eta_{f}}$*)* and $\eta_{f}=\eta_{w}\left(1+2.5C\right)$; $\rho_{\mathrm{mix}}$, $\rho_{f}$, and $\rho_{w}$ are the density of fluid mixture, ferrofluid, and water, respectively; $\eta_{\mathrm{mix}}$, $\eta_{f}$, and $\eta_{w}$ are the dynamic viscosity of fluid mixture, ferrofluid, and water, respectively; *C* is the ferrofluid concentration.

For the cases with externally imposed magnets, the magnetic field intensity **H**, is calculated first by solving for the Maxwell equations using the FEMM software (Version 4.2) and the magnetic flux density field B is calculated following the basic relation of $B=\mu_{0}\left(1+\chi_{f}\right)H$, where $\mu_{0\;}=4\pi \times 10^{-7}\;\frac{H}{m}\;$ is the vacuum permeability, and $\chi_{f}=0.5\;$is the magnetic susceptibility of original ferrofluid.

In this paper, all the magnetic strength fields discussed above are those of the remnant flux density which is a property of the fabricated microscale magnet. Based on the simulated magnetic field in the microchannel, the magnetic forces that act on the ferrofluid in the channel length and channel width directions are calculated following Equations (6) and (7) below depending on the local ferrofluid concentration [27,31].
(6)$$F_{m,x}=\frac{C\;\chi_{f}}{\mu_{0}\mu_{r}^{2}}\;\left(\frac{\partial A_{z}}{\partial y}\frac{\partial^{2}A_{z}}{\partial x\partial y}+\frac{\partial A_{z}}{\partial x}\frac{\partial^{2}A_{z}}{\partial x^{2}}\right)$$
(7)$$F_{m,y}=-\frac{C\;\chi_{f}}{\mu_{0}\mu_{r}^{2}}\;\left(\frac{\partial A_{z}}{\partial x}\frac{\partial^{2}A_{z}}{\partial x\partial y}+\frac{\partial A_{z}}{\partial y}\frac{\partial^{2}A_{z}}{\partial y^{2}}\right)$$
where *A* is the magnetic potential that satisfies the relationship with the magnetic flux density field of B=∇×A, where$\;\mu_{r}$ is the relative permeability related to the magnetic susceptibility as $\mu_{r}=1+\chi_{f}$.

The modeling workflow in the current study consists of two steps: the magnetic fields distributions generated by the arrays of magnets are solved for first using the FEMM software based on a finite-element framework (Figure 1e); the computed magnetic field is then imported in the commercial CFD package ANSYS Fluent^®^ (Ansys Inc., Canonsburg, PA, USA) for the calculation of the magnetic forces defined in Equations (6) and (7) as source terms in the momentum equations—because of the diluted nano-magnetic particle concentration in the ferrofluid, the induced magnetic field is orders of magnitudes smaller than the external magnetic field, thus negligible. The velocity and pressure fields are computed for the mixture fluid coupling the ferrofluid concentration field solved as a user-defined scalar (UDS) in the transport equation. The solutions of coupled fields are updated in each iteration until they are converged with unscaled residuals below 10^−8^.

Figure 4 shows the computational domain and mesh used in the CFD simulations. A two-dimensional view of the Y-shaped microchannel is illustrated in Figure 4a, and the simulations are carried out on a 2-D domain and mesh as shown in Figure 4b, in which a total of 0.2 million quadrilateral cells are adopted for the mesh. For the boundary conditions, the no-slip wall boundary condition is applied at all the side walls of the microchannel, and the velocity inlet B.C. is used at the two channel entrances for water and ferrofluid respectively, as shown in Figure 4a. The inlet velocities are calculated based on the throughputs of the two fluids and the inlet opening area. At water inlet, the ferrofluid concentration is prescribed as 0, while the value is set to nano-magnetic particle concentration at the entrance of the ferrofluid. At all the side walls of the micromixer, the zero-flux B.C. is used for the ferrofluid concentration UDS. At the microchannel mixer exit, the pressure boundary condition is set with a fixed total absolute pressure value of 1 atm. In this simulation, the water-based ferrofluid is a Newtonian fluid. In the cases of non-Newtonian fluid flow, the software ANSYS Fluent provides several constitutive models (such as power-law model, Carreau model, Herschel–Bulkley model, e.g., for different fluid properties) that can be used in laminar flows. User-defined function can also be developed to achieve different rheology characteristics and plasticity of the non-Newtonian fluids.

## 4. Results and Discussion

The magnetized NdFeB–PDMS microstructure functions as permanent magnets, which exerts magnetic forces on the nano Fe_3_O_4_ particles in ferrofluid and induces the mixing between ferrofluid and buffer flow of distilled water. This section discusses the effect of total flow rate, ferrofluid concentration, and magnet mass ratio of NdFeB:PDMS on the mixing efficiency inside the fluidic channel from experiments. The numerical simulations were performed using ANSYS Fluent to explain the mixing behavior, and the simulation results are validated by experimental measurements. The mixing degree is evaluated using the intensity in the imaged fluid volume at different positions along the microchannel. The degree of mixing ($C_{m}$) is calculated according to the following equation proposed in [32,33]:(8)$$C_{m}=1-\frac{\sqrt{\frac{1}{N}\sum_{i=1}^{N}{\left(X_{i}-\bar{X}\right)}^{2}}}{\bar{X}}$$
where $X_{i}$ is the intensity of each pixel in the cross section extracted from experimental photos by in-house Matlab codes using ImageJ [34]. N is the number of total pixels and $\bar{X}$ is the average intensity of all the pixels. As defined in Equation (8), a larger $C_{m}$ indicates less deviation of the intensities from its average value from the photo, and thus better mixing performance between the two streams of fluids.

### 4.1. Effect of Total Flow Rate on Mixing Efficiency

Figure 5 shows the effect of total flow rate on the mixing efficiency between ferrofluid and distilled water. Ferrofluid with a nano-magnetic particle concentration of 1.2% (v/v) and buffer flow of distilled water were injected into the microfluidic channel from upper and lower inlet respectively under the same volumetric flow rate for different total flow rates. Figure 5(a1) is the fluids distribution at the inlet of microfluidic channel at a total flow rate of 0.3 mL/h. It is evident that the widths of ferrofluid and buffer flow were roughly the same because the flow rate ratio between these two streams is 1:1. However, both experiment observations and CFD predictions suggest a slightly thicker layer of the ferrofluid because of its slightly higher dynamic viscosity compared to water. As can be observed from Figure 1d, the microfluidic channel inlet was still far from the micromagnet, thereby, the magnetic field was too weak to immediately affect the mixing and the interface between two fluids stayed sharp. Similar results were also found for all the other cases where total flow rates ranged from Q = 0.4 mL/h to Q = 1.2 mL/h, that no mixing of the ferrofluid with buffer was observed near the channel inlets and a sharp interface remained since the magnetic field and its gradient were extremely weak (data not shown due to repetition with Figure 5(a1)). Figure 5(a2) shows that the CFD simulation results under the experimental conditions in Figure 5(a1) exhibit an excellent match with the measurements.

Figure 5(b1–b4) demonstrate the mixing performance at different streamwise locations from upstream to the outlet of microfluidic channel when the total flow rate Q is 0.3 mL/h. The sharp interface between the ferrofluid and buffer flow at the inlet gradually became blurry towards downstream. As shown in Figure 5(b4), the interface disappeared at channel exit, which suggests a complete and homogenous mixing between the ferrofluid and distilled water. This is because as the ferrofluid moved along the fluidic channel, the magnetic force acted on the nano magnetic particles suspended in ferrofluid for a longer time, and thus a more complete the mixing performance with buffer flow. Figure 5(b5) is the simulated mixing performance under the same experimental conditions with Figure 5(b4) at the outlet and shows very excellent agreements with experimental results. Figure 5(c1–c4)–(f1–f4) show the mixing performance between ferrofluid and distilled water at different *x* locations for various total flow rates. The mixing process for other total flow rates (0.4–1.2 mL/h) resembles that in the 0.3 mL/h case but differs in the mixing degree at the outlet of the microchannel. It is obvious by qualitative comparison of the experiment observations that increasing total flow rates reduced the mixing degree. In all cases, the numerical simulation results in Figure 5(b5–f5) of fluid mixture concentration near a total mixing length of 20 mm match closely with the experiment measurements and the same trend is exhibited reflecting the effect of total flow rates on the mixing performance. As a comparison experiment, it is shown through Figure 5(g1–g5) that the water and ferrofluid interface persisted near the channel outlet, which clearly separated the fluids into two parts. As expected, mixing is limited by the diffusion between the two fluids, therefore extremely slow without the external magnetic field under such low Reynolds numbers. Comparing Figure 5b–f with Figure 5g at corresponding channel positions, it is clearly observed that ferrofluid under externally imposed micro-magnets can significantly improve the mixing efficiency and overcome the laminar diffusion barrier.

The effects of both the distance from inlet and the total flow rate on the mixing efficiency are reflected from Figure 6a. When the distance from the microfluidic channel inlet increased, the mixing efficiency kept increasing for all the groups with different flow rates. This can be explained by the fact that the larger distance from inlet indicates the longer residence time of mixing fluids in the mixing channel, which thus resulted in a more thorough mixing performance. Furthermore, as illustrated by Figure 6a, the mixing efficiency of a lower flow rate was always higher than that of a higher flow rate at the same streamwise positions of microfluidic mixing channel. The reason is that with an increasing volumetric flow rate and thus increasing average flow velocity of the fluids in the microchannel, the residence time of magnetic force acting on the ferrofluid nanoparticles became shorter. Therefore, the two fluids didn’t have enough time to mix thoroughly. Comparing the mixing performance at x = 20 mm in Figure 5(b4–f4), it is evident that the fluids mixed thoroughly in Figure 5(b4) with Q=0.3 mL/h, while the interfaces still existed in Figure 5e4 Q=0.6 mL/h and Figure 5(f4) Q=1.2 mL/h indicating the mixing hasn’t been completed. This suggests that increasing flow rate will shorten the residence time of mixing fluids in the microfluidic channel, so the two fluids don’t have enough time to be mixed with each other. The finding is more evident in Figure 6b which indicates a clear trend of the effect of the total flow rate on the mixing degree at the outlet of microfluidic channel, which is that as the total flow rate increased, the mixing degree decreased. Although it has been clearly shown that the mixing degree is positively correlated to the residence time of flow, a quantitative correlation can be determined based on the experiment measurements. 

The measured mixing time for the device in the current work is in the range of 300 to 600 ms depending on the tested throughputs. This resultant mixing time is close to or faster than the results published in previous works [26,35] which used similar microchannel scales and ferrofluids for mixing enhancement. Lapidus [36] utilized advanced fabrication techniques and performed mixing experiments in the microchannel with a much smaller scale than in the current work. The mixing time from Lapidus’ experiments reached as low as a few microseconds, which is smaller than that in the current work. This is mainly because of the vastly different scales of the channel sizes used in the two experiments, and the flow configurations in the channel. The objective of the current work is to develop a non-expensive fabrication and operation procedure that suits the high-throughput mixing conditions, and enhance the channel mixing using micro-magnets. Despite operating at different scales for varies application, the channel and flow configurations in Lapidus’ work exemplifies the possibility of further improvement of the designs in the current setup, which could be explored in future work.

### 4.2. Effect of Ferrofluid Concentration on the Mixing Efficiency

In order to investigate the effect of ferrofluid concentration, the mixing performance are compared between 1.2% (v/v) and 0.4% (v/v) water-based ferrofluids with a total flow rate of Q = 0.4 mL/h. It was found in Figure 7 groups (a) and (b) that at the corresponding same x locations, the mixing performance was improved with a higher concentration of ferrofluid. Figure 7(c1,c2) are the simulation results corresponding to the mixing performance at inlet (Figure 7(b1)) and outlet (Figure 7(b5)) of microfluidic channel, respectively, when the concentration of ferrofluid is 0.4% (v/v), and suggest a good match with experiments. The Figure 7d clearly reveals that the 1.2% (v/v) concentration of ferrofluid will result in a better mixing performance between ferrofluid and buffer flow. The initial difference in the mixing degree between the 1.2% and 0.4% concentrations resided in the contrast difference between the two cases, where the more diluted ferrofluid exhibited less deviation of intensity from the average value of all the pixels. For 0.4% (v/v) EMG 408, although mixing progresses in the streamwise direction, the mixing degree at the outlet is still very low, indicating a weaker mixing performance. This can be explained by Equations (6) and (7), where the magnetic forces exerted on the ferrofluid is calculated as proportional to the ferrofluid concentration. Therefore, increasing the concentration of ferrofluid can result in greater magnetic force which in return improves the mixing efficiency, as is observed in the case with 1.2% (v/v) ferrofluid concentration. It is worth noting that the color scheme for the CFD simulation results in Figure 7(c2) showing the mixing status at microchannel exit utilized 1 as the ferrofluid volume fraction in the mixture, instead of the concentration of the nano-magnetic particles in the ferrofluid, therefore deviated from that in the microscopy photo. 

### 4.3. Effect of Micromagnet Mass Ratio of NdFeB:PDMS on the Mixing Efficiency

To examine the effect of microscale magnet strength on the mixing performance, two sets of microscale magnets were fabricated by the mixture of neodymium powders (NdFeB) and PDMS with their mass ratio of 2:1 and 1:1 respectively, and experiments were performed comparing the two under a fixed flow rate of 0.3 mL/h. It was observed from the snapshots in Figure 8a,b that at the same streamwise locations, the mixing degree is higher under the higher NdFeB:PDMS mass ratio of 2:1. Figure 8c reveals that a better mixing performance between ferrofluid and buffer flow can be achieved by the micromagnet with the 2:1 NdFeB:PDMS mass ratio at each tested location along the streamwise direction, with a mixing degree about 30–40% higher than in the other case. This can be explained by examining the magnetic fields generated by the micromagnet with different NdFeB:PDMS mass ratios. As sketched in Figure 8d, a piece of straight line with a length of 5000 µm was drawn in the streamwise direction at the center of the microfluidic channel, along which the magnitude of magnetic flux density |**B**| was plotted from the simulated field by FEMM. It is apparent that the magnitude of the calculated magnetic flux density |**B**| was proportional to the NdFeB:PDMS mass ratio used in the fabrication of the microscale magnets, thus about twice higher when the mass ratio of NdFeB:PDMS was 2:1. The wavy pattern of the |**B**| field profile in Figure 8d reflects the non-uniform magnetic flux density distributions generated by this specific magnet design, from which it is also observed that the gradient of the |**B**| field is again higher with a higher NdFeB:PDMS mass ratio. Therefore, based on Equations (6) and (7), the magnetic forces exerted on the ferrofluid in the 2-D channel plane are much greater given a higher |**B**| field and a higher magnetic gradient under the higher NdFeB:PDMS mass ratio in the fabricated micromagnet. This study therefore suggests that increasing the mass ratio of NdFeB:PDMS in the fabricated micromagnets generates greater magnetic forces to promote the transverse transport of the ferrofluid momentum and enhance the mixing degree considerably. 

## 5. Conclusions

In this paper, we proposed a microfluidic device that can achieve the rapid mixing of ferrofluid and distilled water by utilizing a miniaturized and integrated microscale magnet. To accomplish this, a high-gradient microscale magnet was developed by a simple fabrication technique. The integrated NdFeB-PDMS microscale permanent magnet was fabricated and located on one side of the microchannel, with a distance of 150 µm, to accelerate the mixing process. Rapid mixing of ferrofluid and distilled water using external permanent micromagnet made from neodymium powders is analyzed numerically and experimentally in this study. The microfluidic device containing two inlets and one outlet was fabricated using a soft lithography method. Two major parameters, the total flow rate of fluids, the ferrofluid concentration and micromagnet mass ratio of NdFeB:PDMS were systematically investigated via lab-on-chip experiments and numerical simulations, and their effects on the mixing performance in the microfluidics system were thoroughly discussed. The following conclusions could be drawn from this study:(1)A 2-D steady-state CFD model was developed to simulate laminar flow of fluids and their mixing behavior in a micromixer channel, with and without the effect of externally imposed magnetic field. The magnetic field generated by the array of fabricated magnets were calculated in prior with an open-source package of FEMM, which was then imported in the CFD model for coupled simulations of fluid flow and mixing in the microchannel. With the numerical simulation results matching closely with experimental measurements, this modeling workflow is validated.(2)By decreasing the total flow rate, the residence time increased, and the ferrofluid and distilled water had longer time to mix thoroughly with each other inside the microfluidic channel.(3)As the ferrofluid concentration and the strength of the magnet increased, the mixing efficiency also increased due to the stronger magnetic force. These results show that the mixing in the microfluidic channel can be done with the help of a magnet without increasing the length of the channel.(4)The simple yet powerful technique proposed in this work significantly reduces the size of the integrated device and is obviously less expensive fabrication approach. In the meanwhile, the microscale permanent magnets can also be easily adapted to high throughput systems as shown in Figure 9.

## Figures and Tables

**Figure 1 micromachines-11-00029-f001:**
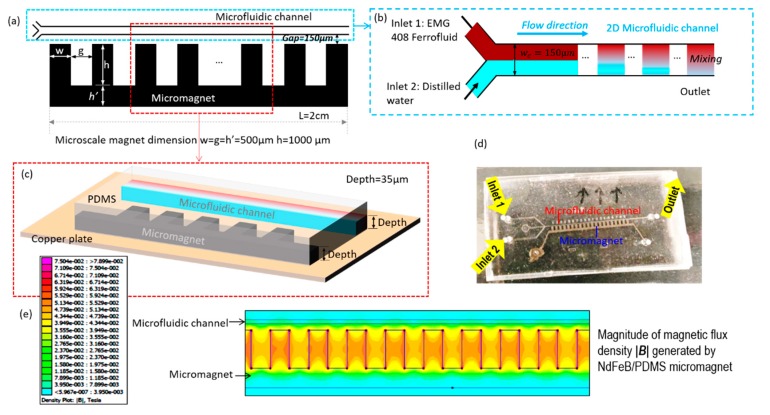
The schematic of the rapid mixing microdevice with embedded microscale magnet. (**a**) The two-dimensional overview of microfluidic channel and micromagnet. The microscale magnet is fabricated beside the microfluidic channel, and the distance between the microscale magnet and the microfluidics channel is gap = 150 µm. The width, length, and gap of the microscale magnet ranges from 500 µm to 1000 µm. (**b**) The enlarged sketch of fluidic channel. The microfluidic channel has a width of $w_{c}$ = 150 µm and the length of the microscale magnet and microfluidic channel are both L=2 cm. Inlet 1 is injected with ferrofluid and Inlet 2 is injected with distilled water. (**c**) The three-dimensional view of sectional microdevice. The depths of the microfluidic channel and microscale magnet are the same with depth = 35 µm which is the thickness of dry photoresist film. (**d**) The prototype of microfluidic device for magnetic rapid mixing. (**e**) The simulated contour of magnetic flux density magnitude |**B**| generated by the micromagnet simulated.

**Figure 2 micromachines-11-00029-f002:**
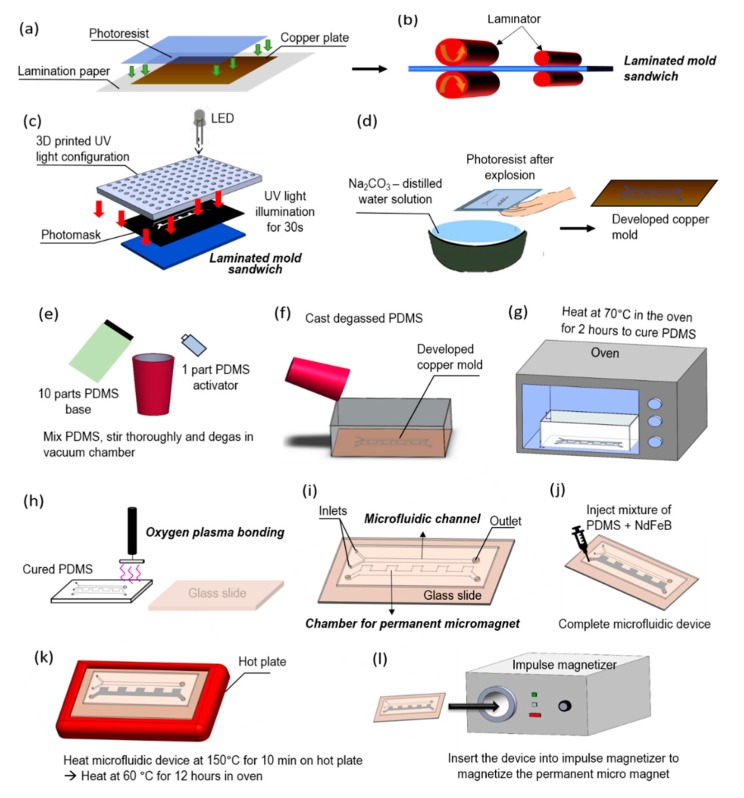
Fabrication steps of the micromixing device with embedded micromagnet. (**a**) A photoresist film, 2 in × 3 in copper plate, and lamination paper are sandwiched together. This configuration will be called “mold sandwich”. (**b**) The mold sandwich is run through a laminator at 100 °C for four times. (**c**) A photomask is placed on the laminated mold sandwich and it is illuminated by custom-made UV light for 30 s. (**d**) The exposed layer of copper plate and photoresist film are developed in a solution of 10 g sodium carbonate powders and 1 L of distilled water for 60 s. (**e**) In a plastic cup, 10 parts of polydimethylsiloxane (PDMS) base and 1 part PDMS activator are mixed together and degassed in a vacuum chamber. (**f**) While the PDMS is degassing, the developed copper plate is placed inside an aluminum box. Once the PDMS solution is totally degassed, it is cast into the aluminum box that contains the copper mold. (**g**) The box with copper mold and liquid PDMS is heated in the oven at 70 °C for 2 h. (**h**) Once the PDMS is cured and cooled down, holes are drilled at the inlets and outlet of the device. A glass slide and the cured PDMS are cleaned thoroughly using isopropanol (IPA) and distilled water. The PDMS and glass slide are then bonded together using oxygen plasma and left on a hot plate at a temperature of 120 °C for 12 h. (**i**) The microfluidic device consisting of channels and chamber for microscale magnet is ready. (**j**) A mixture of neodymium powders and PDMS is injected into the chamber. (**k**) The NdFeB and PDMS mixture is cured on a hot plate and subsequently in the oven. (**l**) The device is inserted into an impulse magnetizer to magnetize the NdFeB-PDMS mixture as a permanent microscale magnet.

**Figure 3 micromachines-11-00029-f003:**
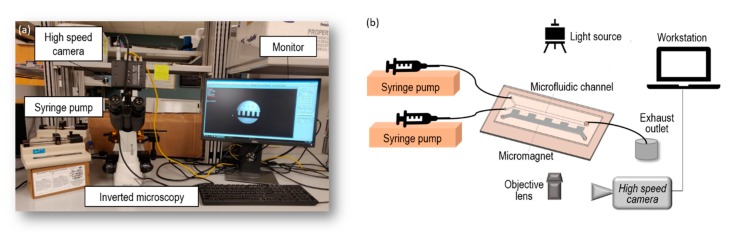
(**a**) Main components of experimental setup: high speed camera, syringe pump, inverted microscopy platform, and computer; (**b**) Schematic of the experimental system for micromixing between ferrofluid and buffer flow.

**Figure 4 micromachines-11-00029-f004:**
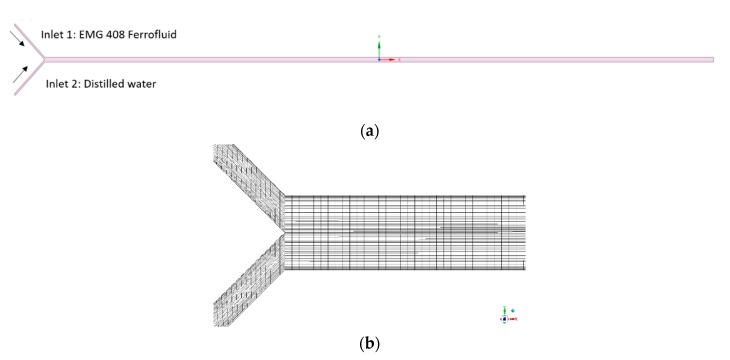
(**a**) Computational domain; (**b**) Mesh near the entrance of microfluidic channel.

**Figure 5 micromachines-11-00029-f005:**
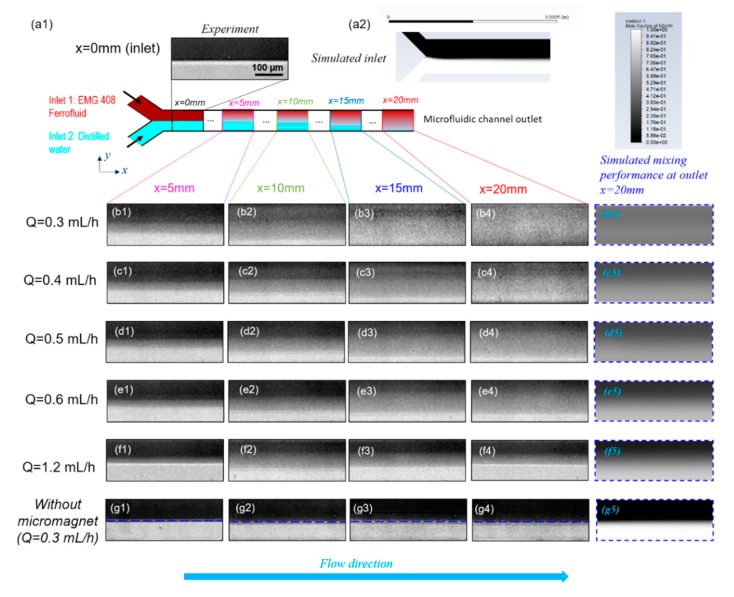
Effect of total flow rate Q on mixing performance. (**a1**) is the experimental distribution of ferrofluid and distilled water. (**a2**) is the corresponding simulation results for mole faction of nano ${\mathrm{Fe}}_{3}O_{4}$ particles in ferrofluid at inlet of microfluidic channel. (**b1**–**b4**) are a group of images at different downstream positions x=5 mm, x=10 mm, x=15 mm, and x=20 mm (outlet) of microfluidic channel when the total flow rate Q=0.3 mL/h. (**b5**) is the simulated mixing performance at outlet. The color represents the mole faction of nano $Fe_{3}O_{4}$ particles in ferrofluid and its legend is the same with that in (**a2**). Groups (**b**)–(**f**) are the corresponding experimental and simulated results for various total flow rate of Q=0.4 mL/h,
Q=0.5 mL/h,
Q=0.6 mL/h, and Q=1.2 mL/h, respectively. Group (**g**) is the experimental and simulated distribution of ferrofluid and distilled water without the effect of magnetic fields when Q=0.3 mL/h. The flow rate ratio between ferrofluid to distilled water is 1:1, and the ferrofluid magnetic nanoparticle concentration is 1.2% (v/v) for all the groups from (**a**–**g**).

**Figure 6 micromachines-11-00029-f006:**
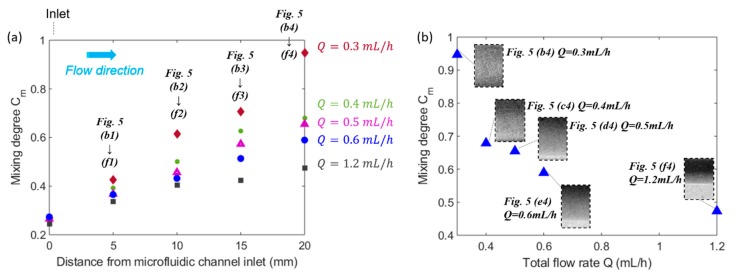
(**a**) The evolution of the mixing degree $C_{m}$ along flow direction for various total flow rate Q. The spots represent the mixing degree corresponding to the experimental images in Figure 5. (**b**) Effect of total flow rate Q on the mixing degree $C_{m}$ corresponding to the images from (b4)–(f4) in Figure 5. The concentration (v/v) of ferrofluid is 1.2% (v/v), the flow rate ratio between ferrofluid to water is 1:1, and the spots represent the mixing performance at the outlet (x = 20 mm) of fluidic channel in Figure 5 for various total flow rates.

**Figure 7 micromachines-11-00029-f007:**
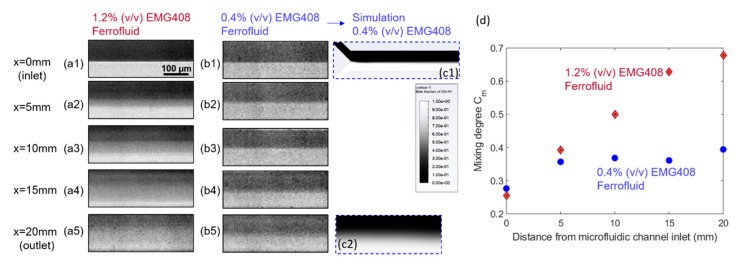
Effect of the concentration of ferrofluid on the mixing performance. Group (**a**,**b**) are the original experimental fluids distribution at different positions of the microfluidic channel under the effect of magnetic field generated by embedded micromagnet when the magnetic nanoparticles concentration of EMG 408 ferrofluid is 1.2% (v/v) and 0.4% (v/v). The total flow rate is 0.4 mL/h and the flow rate ratio between ferrofluid to water is 1:1 for both groups of (**a**,**b**). In (**c**), (**c1**,**c2**) are the simulated results corresponding to (**b1**,**b5**), respectively. (**d**) is the effect of ferrofluid concentration on mixing degree corresponding to group (**a**,**b**).

**Figure 8 micromachines-11-00029-f008:**
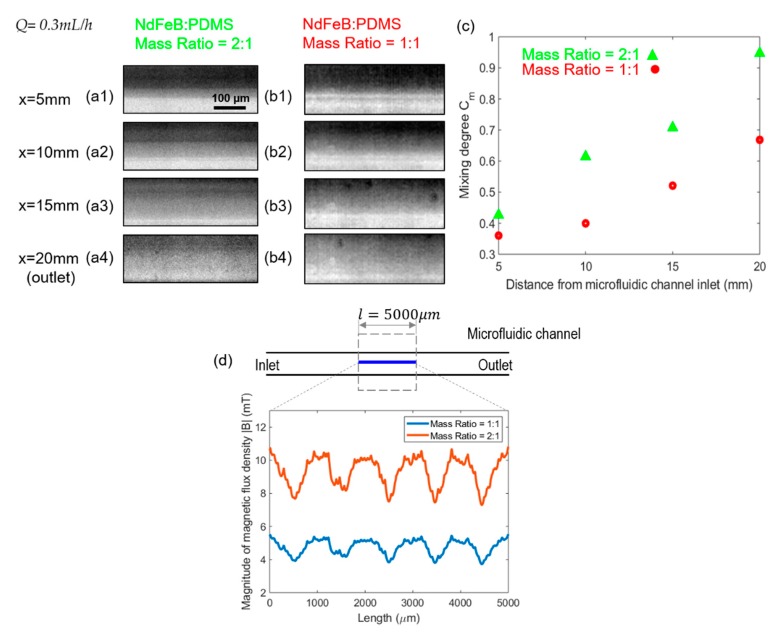
(**a**) Effect of the mass ratio of NdFeB:PDMS micromagnet on the mixing performance. Group (**a**,**b**) are the original experimental fluids distribution at different positions of the microfluidic channel under the effect of magnetic field generated by the embedded micromagnet when its mass ratio of NdFeB:PDMS is 2:1 and 1:1. The total flow rate is 0.3 mL/h and the flow rate ratio between ferrofluid to water is 1:1 for both groups of (**a**,**b**). (**c**) is the effect of mass ratio of NdFeB:PDMS micromagnet on mixing degree corresponding to group (**a**,**b**). (**d**) is the magnitude of magnetic flux density |**B**| generated by the microscale magnet whose mass ratio of NdFeB:PDMS is 2:1 and 1:1 respectively.

**Figure 9 micromachines-11-00029-f009:**
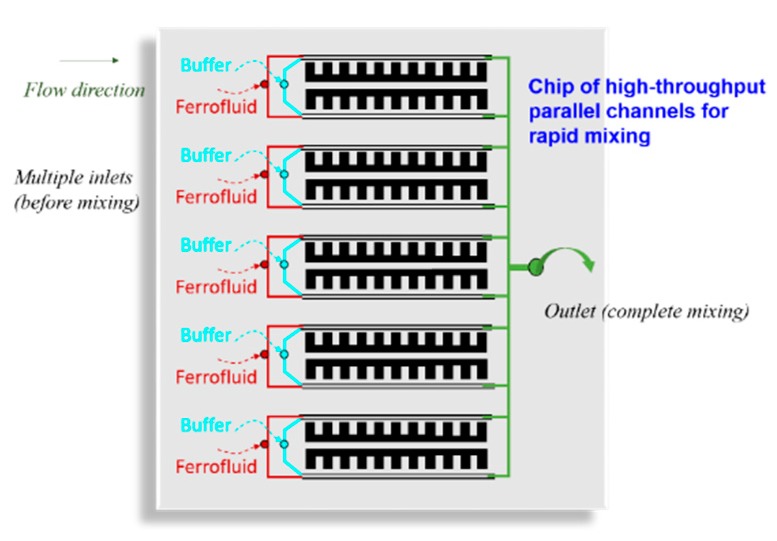
High-throughput microfluidic chip with integrated parallel channels for rapid mixing.

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
