# Peer review of "Rapid Microfluidic Mixer Based on Ferrofluid and Integrated Microscale NdFeB-PDMS Magnet"

_micromachines, 2019, doi:10.3390/mi11010029_

Round 1
Reviewer 1 Report
In this work, the authors demonstrate a microfluidic mixer based on ferrofluid and integrated microscale magnet. Although the results shown in the manuscript are clearly presented and scientifically sound, authors need to address some key points before I can recommend its publication.
1) I am quite confused about goal of this work. Here, the authors test the mixing performance between ferrofluid and water and I agree that there is mixing enhancement using the proposed mixer. However, in order to claim that the mixer can generally enhance the mixing, shouldn't they test the mixing between ferrofluid, water and a third liquid? If the authors apply the current mixer to a real application (i.e. biological application), then it will be the mixing between two components (i.e. one protein solution and one DNA solution). The ferrofluid is just an agent that helps the mixing. Please provide experimental results that the current setup is actually useful in enhancing the mixing between two different liquids besides ferrofluid.
2) In the introduction, the authors write "Ferrofluids have received much attention recently to enhance the mixing efficiency between other sample fluids due to its characteristics of biological compatibility". Do the authors have a reference for this?
3) Do the authors have results from control experiments without magnetic field applied? I think they should provide the results for at least one case (0.3 mL/h) without the magnet.
4) How does the viscosity affect the mixing performance?
5) There are several typos in the manuscript. For instance, applcaiton (line 33), fabricaiton (line 81), magentic (line 86). Please go through the entire manuscript carefully and fix the typos.
Reviewer 2 Report
This manuscript describes a simple lab on a chip set-up using for mixing. The simulation is also provided. A revision is needed before further consideration.
What is the mixing time of the device in this manuscript? and how fast it is compared to other available devices such as this ultra-rapid mixer: https://www.mdpi.com/2072-666X/8/1/16 (Micromachines 2017, 8(1), 16; https://doi.org/10.3390/mi8010016). A discussion on this aspect can be helpful for the readers and should be added into the manuscript. Can the authors include the simulation file in the supplementary material? Is the model in the simulation for Newtonian fluid? In the introduction, the authors mentioned a lot on the biology samples, how can this model be used with biology samples which may be non-Newtonian fluid?Author Response
Please see the attachment.

Round 2
Reviewer 1 Report
I am satisfied with the revised manuscript, which I believe is ready for publication.
Reviewer 2 Report
The authors have addressed my comments. I do not have further comments.